# AGREEMENT WITH THE ENSEMBLE FOR ZERO-SHOT VISION-LANGUAGE MODEL SELECTION

## ABSTRACT

Pretrained vision-language models (VLMs) such as CLIP are well known for enabling *zero-shot* classification with *category names*. The rapid growth of open-access variants has led to a diverse VLM zoo, where selecting the most suitable model can yield superior zero-shot performance, yet the optimal choice is often *dataset-dependent*. At the same time, selecting VLMs for *zero-shot* tasks is challenging, since only *category names* are available and target images are absent. Prior approaches rely on text-only evaluation, which suffers from the *modality gap* inherent to VLMs. To address this issue, we propose **SAGE** (Selection via AGreement-with-the-Ensemble), which leverages *in-the-wild* images to bridge the modality gap. Specifically, SAGE quantifies the agreement between individual VLMs and their ensemble counterparts in terms of prediction behavior on in-the-wild images. Experiments demonstrate that SAGE consistently outperforms state-of-the-art zero-shot VLM selection methods.

## 1 INTRODUCTION

Vision-language models (VLMs) have reshaped the intersection of computer vision and natural language processing by bridging visual and textual modalities (Li et al., 2022; Singh et al., 2022; Wang et al., 2023). A prominent example is CLIP (Radford et al., 2021), which learns aligned image–text representations via contrastive training. One of its most notable capabilities is *zero-shot* image recognition, enabling predictions for unseen image classes using only category names.

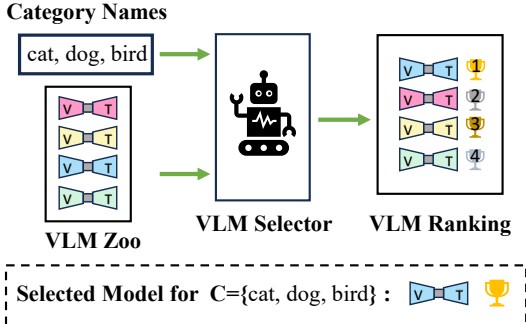

Figure 1: **Zero-shot VLM selection.** The model selector is required to select an appropriate VLM according to category names.

Today, even users without machine learning expertise can download VLMs from the open-source community to perform their own recognition tasks. Within this community, a wide variety of VLMs trained with diverse architectures and strategies form an ever-expanding "VLM zoo." (Zohar et al., 2023; Lu et al., 2024; Jeong et al., 2024) Prior studies show that the zero-shot recognition performance of VLMs is highly *dataset-dependent* (Fang et al., 2022; Rodriguez-Opazo et al., 2025). Consequently, selecting VLMs from the zoo according to the target task with an appropriate strategy can yield better results than arbitrary choice. However, this is particularly difficult for non-expert users, who often lack the necessary experience and the time to collect an evaluation dataset. In many cases, they only have access to the category names. To address these challenges, recent work (Zohar et al., 2023) introduces the task of **zero-shot VLM selection**, which evaluates models solely based on the provided **category names**, without extra target data (see Figure 1).

Existing approaches (Zohar et al., 2023; Yi et al., 2024) rely on *text-only* proxy evaluation. Specifically, LLMs (Ouyang et al., 2022; Touvron et al., 2023) are prompted with category names to generate task-related texts, which are then fed into the text encoders of VLMs to approximate target image features within the cross-modal embedding space. The image recognition capability of VLMs is

Figure 2: **Compare methods of zero-shot VLM selection.** These methods compute a metric $\mu$ on proxy data to estimate test accuracy. *(a) Previous text-only approaches* rely on LLM-generated texts to simulate target images, but the text classification ability does not faithfully reflect the image recognition capability of VLMs due to the modality gap phenomenon. *(b) Our proposed SAGE* avoid the modality gap by leveraging in-the-wild images.

subsequently estimated from the classification performance of text classifiers over these generated textual features. However, such methods attempt to assess cross-modal characteristics using *uni-modal* data, thereby suffering from the well-known *modality gap* (Liang et al., 2022), which creates a discrepancy between discriminative abilities over text and over images.

In this paper, we address the limitation of *uni-modal* evaluation by leveraging **in-the-wild images**, which have been shown to help bridge the modality gap in zero-shot tasks. The difficulty of utilizing in-the-wild images for VLM selection arises from two perspectives: (1) the absence of target task labels for in-the-wild images; and (2) the semantic gap between in-the-wild images and the target task.

To tackle the first challenge, we propose pseudo-labeling with VLM ensembles. Since all models must be inferred during selection, their predictions can be aggregated to form an ensemble model. This ensemble exhibits strong generalization across diverse tasks and thus provides high-quality pseudo-labels. Building on this, we propose **AGE** (agreement-with-the-ensemble), a label-free metric that measures the consistency between individual VLM predictions and pseudo-labels generated by ensembles. We empirically demonstrate that AGE is strongly correlated with accuracy.

To tackle the second challenge, we integrate semantic-based image retrieval with robust estimation to mitigate the semantic gap between in-the-wild images and the target task when computing AGE. Consequently, AGE can be used to estimate accuracy without requiring target images, enabling SAGE to effectively bridge the modality gap in zero-shot VLM selection and achieve state-of-the-art performance. A comparison between our approach and existing methods is shown in Figure 2.

Our contributions are as follows:

- We introduce AGE, a metric that quantifies agreement between individual outputs and their ensemble counterparts, which strongly correlates with the ground-truth accuracy.

- We adapt AGE to zero-shot VLM selection with in-the-wild images, leading to our proposed method SAGE.

- We evaluate SAGE on the established benchmark for zero-shot VLM selection, demonstrating that SAGE outperforms existing SOTA significantly.

## 2 RELATED WORK

**Vision-language models.** Pretrained vision-language models (Li et al., 2022; Singh et al., 2022; Wang et al., 2023), are typically trained on large datasets of image-text pairs (Lin et al., 2014; Young et al., 2014; Schuhmann et al., 2021) and are utilized for a variety of challenging tasks. CLIP (Radford et al., 2021), for example, is designed to create a unified representation space for images and texts, enabling image classification without requiring specific training samples for the target task. This approach, known as "zero-shot classification", maps test images to class names in the learned representation space. Numerous pretrained models have been developed and made

available online for users to download based on their specific tasks. For instance, different versions of CLIP with various architectures (He et al., 2016b; Vaswani et al., 2017; Liu et al., 2022) trained on diverse datasets (Deng et al., 2009; Lin et al., 2014; Schuhmann et al., 2021) can be found on model platforms. These diverse VLMs form a **VLM zoo**. Recent studies (Lu et al., 2024; Jeong et al., 2024) have shown that combining the outputs of multiple VLMs, forming **VLM ensembles**, can achieve higher zero-shot performance.

**Pretrained model selection.** Pretrained models make it possible to utilize prior knowledge learned from a variety of datasets, showing remarkable improvement compared with training from scratch (He et al., 2016a; Devlin, 2018; Radford et al., 2021; Amos et al., 2024). Observing that randomly selecting a pretrained model from multiple models can result in unpredictable performance (Wang et al., 2019), pretrained model selection methods estimate model performance given a zoo of pretrained models and a small set of labeled data. Typically, forward-based methods (Tran et al., 2019; 2020; You et al., 2021; 2022; Ding et al., 2022; Huang et al., 2022a) assess feature-label distribution, while similarity-based methods (Achille et al., 2019; Dwivedi & Roig, 2019; Zhang et al., 2023) extract or learn representations for models and datasets and match models to datasets based on similarity. At the same time, model selection using only unlabeled data has drawn attention recently (Lin et al., 2020; Zhao et al., 2021; Baek et al., 2022; Goswami et al., 2022; Hu et al., 2023).

Unlike other pretrained vision models, vision-language models are often used in zero-shot settings without access to image data, making evaluation challenging. **Zero-shot VLM selection** (Zohar et al., 2023; Yi et al., 2024) addresses the problem of selecting VLMs using only category names.

## 3 Preliminaries

### 3.1 Vision-Language Model

VLMs are known for their cross-modal generalization ability (Radford et al., 2021; Li et al., 2022; Wang et al., 2023; Singh et al., 2022). A VLM $f$ consists of an image encoder $f_{\mathbf{x}}$ and a text encoder $f_{\mathbf{t}}$. Mapping texts and images into a unified representation space, similarity $\text{sim}(\mathbf{x}, \mathbf{t})$ between an image $\mathbf{x}$ and a text $\mathbf{t}$ is obtained by cosine similarity:

$$\text{sim}(\mathbf{x}, \mathbf{t}; f) = \frac{f_{\mathbf{x}}(\mathbf{x})^{\top} f_{\mathbf{t}}(\mathbf{t})}{\|f_{\mathbf{x}}(\mathbf{x})\|_2 \cdot \|f_{\mathbf{t}}(\mathbf{t})\|_2}. \tag{1}$$

**Zero-Shot Classification.** A zero-shot image classification task is defined as $\langle \mathcal{C}, \mathcal{X}_{\text{test}} \rangle$, where $\mathcal{C}$ are **category names** and $\mathcal{X}_{\text{test}}$ are the **test images** to be classified. Specifically, the user describes a $C$-class custom task by providing category names $\mathcal{C} = \{\mathbf{c}_i\}_{i=1}^{C}$ (*e.g.*, {cat, dog, bird}). Then, the category names are plugged into a defined prompt template (*e.g.*, "a photo of a {c}"), forming textual prompts $\mathcal{T} = \{\mathbf{t}_i\}_{i=1}^{C}$. For a test image $\mathbf{x} \in \mathcal{X}_{\text{test}}$, a VLM $f$ calculates similarity between the test image and the prompts of each class. Then the predicted probability of class $y$ is given by:

$$\Pr(\hat{y} = y | \mathbf{x}, \mathcal{T}, f) = \frac{\exp\left(\text{sim}(\mathbf{x}, \mathbf{t}_y; f)\right)}{\sum_{y' \in [C]} \exp\left(\text{sim}(\mathbf{x}, \mathbf{t}_{y'}; f)\right)}. \tag{2}$$

We use $f(\mathbf{x}; \mathcal{T}) \in \Delta^C$ to denote the probability output of model $f$ given image $\mathbf{x}$ and class prompts $\mathcal{T}$, where $f(\mathbf{x}; \mathcal{T})_i \triangleq \Pr(\hat{y} = i | \mathbf{x}, \mathcal{T}, f)$. The predicted label would be the class with highest probability. With ground-truth labels, we can measure the *zero-shot* accuracy of model $f$ for a given task $\langle \mathcal{C}, \mathcal{X}_{\text{test}} \rangle$ with corresponding labels $\mathcal{Y}_{\text{test}}$, denoted $\text{acc}(f; \mathcal{C}, \mathcal{X}_{\text{test}}, \mathcal{Y}_{\text{test}})$.

### 3.2 Zero-Shot VLM Selection and Current Solutions

Nowadays, many VLMs are trained and released, varying in terms of the training dataset, model architecture, training methodology, etc.These diverse VLMs form a model zoo, consisting of $M$ VLMs, denoted as $\mathcal{F} = \{f_i\}_{i=1}^{M}$. For a downstream task $\langle \mathcal{C}, \mathcal{X}_{\text{test}} \rangle$, we want to deploy the best model $f^* \in \mathcal{F}$ such that $\text{acc}(f^*; \mathcal{C}, \mathcal{X}_{\text{test}}, \mathcal{Y}_{\text{test}})$ is maximized.

**Problem setting of zero-shot VLM selection.** In this paper, we tackle the challenge of *zero-shot* VLM selection, which requires choosing an optimal VLM from $\mathcal{F}$ according to $\mathcal{C}$ without access to target images $\mathcal{X}_{\text{test}}$. To address the absence of target evaluation data, prior work leverages *proxy*

*data* to compute a *proxy metric* $\mu(f; \mathcal{C})$ that is expected to correlate with the ground-truth accuracy $\text{acc}(f; \mathcal{C}, \mathcal{X}_{\text{test}})$. Such proxy data may consist of **in-the-wild images** from large-scale datasets (*e.g.*, ImageNet (Deng et al., 2009)) or **LLM-generated texts** (Ouyang et al., 2022; Touvron et al., 2023) derived from $\mathcal{C}$. In what follows, we introduce two representative proxy metrics built upon these two directions.

**ImageNet accuracy.** A straightforward baseline is to leverage benchmark performance (*e.g.*, accuracy on ImageNet) as a measure of a VLM's inherent capability:

$$\mu_{\text{IN}}(f; \mathcal{C}) = \text{acc}(f; \mathcal{C}_{\text{IN}}, \mathcal{X}_{\text{IN}}, \mathcal{Y}_{\text{IN}}), \tag{3}$$

where $\mathcal{C}_{\text{IN}}$ and $\mathcal{X}_{\text{IN}}$ are the category names and the test images of ImageNet, respectively.

**Text scores.** ModelGPT (Zohar et al., 2023) and its improved derivative SWAB (Yi et al., 2024) use LLMs to generate captions for each class based on $\mathcal{C}$, forming a "text dataset" $\text{LLM}(\mathcal{C})$. Specifically, given the category names of the target task, an LLM is prompted to produce relevant captions. For example, for a task with categories *Abyssinian* and *Beagle*, it may generate "An adorable Abyssinian cat lounged in sunshine, eyes gleaming afar." for *Abyssinian*, and "A stunning beagle sat on the grass, gazing into distance." for *Beagle*. Based on the cross-modal embedding space, we can treat the generated captions as target images in the cross-modal embedding spaces of VLMs and convert VLM evaluation to assessment of text classification capability, with text classification accuracy treated as an approximation of zero-shot classification accuracy:

$$\mu_{\text{text}}(f; \mathcal{C}) = \text{acc}(f; \mathcal{C}, \text{LLM}(\mathcal{C})). \tag{4}$$

Beyond text classification accuracy, Zohar et al. (2023) propose using more text-based scores such as F1-score and Fisher Criterion, which are combined with ImageNet accuracy to build a strong proxy metric. However, text scores suffer from limited assessment of the visual modality due to the *modality gap* (Liang et al., 2022). Moreover, employing LLMs introduces additional computational cost and makes selection performance highly dependent on the quality of LLM outputs.

# 4 METHODOLOGY OF SAGE

**High-level idea of SAGE.** Our goal is to estimate $\text{acc}(f^*; \mathcal{C}, \mathcal{X}_{\text{test}}, \mathcal{Y}_{\text{test}})$. The challenge is that only $\mathcal{C}$ is available, while the target data $\mathcal{X}_{\text{test}}$ and labels $\mathcal{Y}_{\text{test}}$ are not. To address this, we propose two sequential steps: **(1)** replace ground-truth labels $\mathcal{Y}_{\text{test}}$ with *pseudo-labels* $\widehat{\mathcal{Y}}$, and **(2)** replace target images $\mathcal{X}_{\text{test}}$ with *in-the-wild* images $\mathcal{X}'$.

The first step leads to the **AGE** (agreement-with-the-ensemble) metric, which leverages VLM ensembles to generate *pseudo-labels* for evaluation. The second step adapts AGE to in-the-wild images, where we incorporate *semantic retrieval* and *robust similarity measures* to strengthen this adaptation. In what follows, we formally introduce AGE and demonstrate its correlation with accuracy, and then detail how AGE can be adapted to in-the-wild images for zero-shot VLM selection.

## 4.1 AGE ON TARGET IMAGES

We first tackle the challenge of absence of ground-truth labels. Building on the universal generalization ability of VLMs (Mayilvahanan et al., 2024; Bielawski et al., 2022; Tu et al., 2023) and advances in ensemble learning with VLMs (Lu et al., 2024; Jeong et al., 2024; Li et al., 2023; Huang et al., 2022b), we argue that VLM ensembles provide high-quality pseudo-labels for unlabeled images. Building on this, we propose **AGE** (agreement-with-the-ensemble), which measures the consistency between individual VLMs and their ensemble counterparts.

**Formal definition of AGE.** Consider zero-shot classification on an image set $\mathcal{X}_{\text{test}}$ using VLMs $\mathcal{F} = \{f_i\}_{i=1}^{M}$ and a set of category names. For category names $\mathcal{C}$ consisting of $C$ classes, we insert them into a prompt template to obtain textual prompts $\mathcal{T}$, which are then fed into the text encoders of VLMs to construct text classifiers. For an image $\mathbf{x}$, each VLM $f_i$ outputs a probability vector $f_i(\mathbf{x}; \mathcal{T}) = [p_1, \ldots, p_C] \in \Delta^C$. We then construct an **ensemble model** $\bar{f}$, which produces predictions

$$\bar{f}(\mathbf{x}; \mathcal{T}) \triangleq \frac{1}{M} \sum_{i=1}^{M} f_i(\mathbf{x}; \mathcal{T}). \tag{5}$$

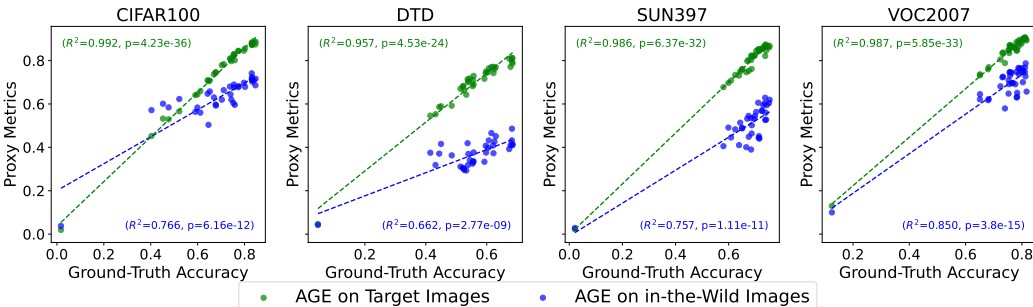

Figure 3: **Linear correlation between AGE and ground-truth accuracy.** Each point represents $(\mathrm{acc}(f_i), \mu(f_i))$ for a VLM $f_i \in \mathcal{F}$. The X-axis denotes ground-truth accuracy, while the Y-axis denotes AGE obtained using target images (green) and in-the-wild images (blue) with pseudo-labels generated by the ensemble model.

Next, for each model $f_i$, we compute AGE $\mu$ on $\mathcal{X}$. AGE is defined as the pseudo-label accuracy of individual models with respect to the pseudo-labels produced by $\bar{f}$:

$$\mu(f_i; \mathcal{C}, \mathcal{X}_{\text{test}}) = \frac{1}{|\mathcal{X}_{\text{test}}|} \sum_{\mathbf{x} \in \mathcal{X}_{\text{test}}} \mathbb{I}\left( \arg\max_j f_i(\mathbf{x}; \mathcal{T})_j = \arg\max_j \bar{f}(\mathbf{x}; \mathcal{T})_j \right). \quad (6)$$

**Observation: AGE strongly correlates with ground-truth accuracy.** We calculate the zero-shot accuracy $\mathrm{acc}(f_i)$ and the AGE $\mu(f_i)$ for the 35 models in the model zoo (see section 5) on selected datasets. We plot $(\mathrm{acc}(f_i), \mu(f_i))$ in green in Figure 3. As shown in Figure 3, AGE exhibits a strong correlation with ground-truth accuracy. Note that all models must be inferred during model selection; therefore, constructing the ensemble model does not significantly increase the time cost. Consequently, AGE can be used for model selection when target labels are absent.

## 4.2 ADAPTING AGE FOR ZERO-SHOT VLM SELECTION

We have demonstrated that when unlabeled target images and category names are available, AGE can be computed and shows strong correlation with ground-truth accuracy, enabling effective VLM ranking and selection without labeled data. However, target images are unavailable for zero-shot VLM selection. In this section, we show that AGE can still serve as a reliable model selection metric even *without* target images.

**Utilizing in-the-wild images.** To address the absence of target images, we propose utilizing in-the-wild images as surrogates. Prior studies (Shin et al., 2022; Wallingford et al., 2023) have shown that such images help bridge the modality gap in zero-shot applications. For broad applicability, the selected in-the-wild images should cover diverse domains to ensure adaptability across downstream tasks. In this work, we adopt ImageNet (Deng et al., 2009) as the source of in-the-wild images. ImageNet is both comprehensive and known to approximate the training distribution of many VLMs (Shin et al., 2022). Specifically, we randomly sample one image from each ImageNet1K class to form a compact in-the-wild dataset for efficiency, denoted as $\mathcal{X}'$.

**Semantic retrieval for task customization.** While in-the-wild images provide general coverage, they may not align semantically with a specific downstream task, creating a *semantic gap*. Intuitively, images semantically closer to the task should contribute more. To this end, we apply a **semantic retrieval** strategy: images are weighted by their similarity to the task category names in the embedding space. Since $\mathcal{X}'$ is small, retrieving only a few samples risks high variance; instead, we employ *soft retrieval*, assigning weights based on semantic similarity and aggregate the results. Formally, each image $\mathbf{x} \in \mathcal{X}'$ is assigned weight

$$w(\mathbf{x}) = \frac{\exp\left( \sum_{\mathbf{t} \in \mathcal{T}} \mathrm{sim}(\mathbf{x}, \mathbf{t}; f_0)/C \right)}{\sum_{\mathbf{x} \sim \mathcal{X}'} \exp\left( \sum_{\mathbf{t} \in \mathcal{T}} \mathrm{sim}(\mathbf{x}, \mathbf{t}; f_0)/C \right)}, \quad (7)$$

where $f_0$ is any VLM from the model zoo. Adapting Equation 6 with in-the-wild images and semantic retrieval, we compute

$$\mu(f_i; \mathcal{C}) = \sum_{\mathbf{x} \sim \mathcal{X}'} w(\mathbf{x}) \cdot \mathbb{I}\big( \arg \max_j f_i(\mathbf{x}; \mathcal{T})_j = \arg \max_j \bar{f}(\mathbf{x}; \mathcal{T})_j \big) . \quad (8)$$

As shown in Figure 3 (blue points), AGE remains correlated with ground-truth accuracy even when target images are replaced by ImageNet alternatives. Although this substitution introduces approximation error, reflected in the weaker correlation compared to using target images (green points), it still preserves the relative model rankings effectively.

**Robust estimation of AGE.** Both Equation 6 and Equation 8 are based on *Top-1 accuracy*, which is well known to be non-robust under a *small* sample size. Moreover, working with in-the-wild data often leads to *low confidence*, further reducing the reliability of accuracy metrics. To address this, we explore alternative, more robust evaluation metrics to support AGE in the in-the-wild setting. In the following, we introduce additional **similarity measures** to quantify the consistency between predictions of individual VLMs and pseudo-labels, *i.e.*, consistency between $f_i(\mathbf{x}; \mathcal{T})$ and $\bar{f}(\mathbf{x}; \mathcal{T})$, which are both probabilistic vectors in the simplex $\Delta^C$.

*(1) Class ranking correlation.* A natural way to compare class probability vectors is to treat them as class rankings and measure the correlation between their induced orderings. Inspired by recommendation systems, we focus on mutual items in the Top-$k$ predictions to highlight high-probability classes, which are more relevant to the target and its common confusions:

$$\text{Sim}(f_i(\mathbf{x}; \mathcal{T}), \bar{f}(\mathbf{x}; \mathcal{T})) = \frac{|\text{argmax}_k(f_i(\mathbf{x}; \mathcal{T})) \cap \text{argmax}_k(\bar{f}(\mathbf{x}; \mathcal{T}))|}{k}, \quad (9)$$

where $\text{argmax}_k$ denotes the indexes of the Top-$k$ components. When $k$ equals 1, the score $\text{Sim}(f_i(\mathbf{x}; \mathcal{T}), \bar{f}(\mathbf{x}; \mathcal{T}))$ reduces to vanilla AGE. In SAGE, we set $k \in \{1, 2, 3\}$.

*(2) Exponential of negative divergence.* A common way to measure the discrepancy between two distributions is via KL divergence. To improve interpretability, we apply the negative exponential to map it into $[0, 1]$, yielding a similarity score:

$$\text{Sim}(f_i(\mathbf{x}; \mathcal{T}), \bar{f}(\mathbf{x}; \mathcal{T})) = \exp \big( - \text{KL}(f_i(\mathbf{x}; \mathcal{T}) || \bar{f}(\mathbf{x}; \mathcal{T})) \big). \quad (10)$$

*(3) Normalized total variation distance.* Total variation distance is another popular way to measure distributional discrepancy, enjoying numerical stability. We normalize it to $[0, 1]$ to obtain a similarity function:

$$\text{Sim}(f_i(\mathbf{x}; \mathcal{T}), \bar{f}(\mathbf{x}; \mathcal{T})) = 1 - \frac{\|f_i(\mathbf{x}; \mathcal{T}) - \bar{f}(\mathbf{x}; \mathcal{T})\|_1}{2}. \quad (11)$$

We replace $\mathbb{I}\big(\text{argmax} f_i(\mathbf{x}; \mathcal{T}) = \text{argmax} \bar{f}(\mathbf{x}; \mathcal{T})\big)$ in Equation 8 with $\text{Sim}(f_i(\mathbf{x}; \mathcal{T}), \bar{f}(\mathbf{x}; \mathcal{T}))$ to compute AGE on $\mathcal{X}'$. We observe that combining multiple implementations of the similarity function yields better performance than relying solely on accuracy-based implementations.

## 5 EXPERIMENTS

In this section, we provide numerical results of our zero-shot VLM selection method.

**Model zoo and datasets.** To evaluate the effectiveness of VLM selection methods, we construct a VLM zoo and test across diverse datasets. Following Zohar et al. (2023), we build a VLM zoo of 35 models from Ilharco et al. (2021), covering variations in architecture, training data, and optimization strategies. We then evaluate selection methods on 23 downstream datasets used in Zohar et al. (2023), including Stanford Cars (Krause et al., 2013), CIFAR-100 (Krizhevsky, 2009), among others. These datasets span a broad range of tasks and domains, providing a comprehensive benchmark. Detailed descriptions of the models and datasets are provided in the appendix.

**Competitive methods.** We compare against three existing approaches: the ImageNet accuracy baseline, ModelGPT (Zohar et al., 2023), and the recent SWAB method (Yi et al., 2024), all of which represent state-of-the-art zero-shot VLM selection techniques.

**Metrics.** For model ranking, we adopt standard evaluation metrics including *Recall@K* and *Weighted Kendall's* $\tau$. In addition, we report the *accuracy* of the selected models to directly assess the effectiveness of model selection.

Table 1: Results on zero-shot VLM selection benchmark

| Method | Use of LLM | $R_5$ | $\tau$ | $R_5 + \tau$ |
|---|---|---|---|---|
| ImageNet Accuracy | $\times$ | 48.7 | 24.6 | 73.3 |
| ModelGPT (Zohar et al., 2023) | $\checkmark$ | 49.6 | 29.0 | 75.6 |
| SWAB (Yi et al., 2024) | $\checkmark$ | 49.8 | 31.0 | 80.8 |
| SAGE | $\times$ | 54.8 | 31.6 | 86.4 |
| SAGE + ModelGPT | $\checkmark$ | **57.4** | **41.4** | **98.8** |

### 5.1 Zero-Shot VLM Selection Benchmark Results

**Protocol.** Following Zohar et al. (2023), we partition the downstream datasets into one *target dataset* and the remaining *support datasets*. VLM selection methods have *full access* to the support datasets, including category names and test accuracies, while only *category names* are available for the target dataset. In addition, methods may exploit external *in-the-wild* data to aid model selection; following prior work, we use ImageNet1K (Deng et al., 2009) for this purpose. To evaluate performance, we adopt a leave-one-out strategy: in each round, one dataset is designated as the target while the others serve as support. The selection methods are required to predict a model ranking for the target dataset, which is then compared against the ground-truth ranking. Following Zohar et al. (2023); Yi et al. (2024), we report **Recall@5** and **Weighted Kendall's** $\tau$, where the weights are assigned uniformly to the mutual Top-5 items.

**Implementation details.** As in ModelGPT and SWAB, SAGE involves combining multiple proxy metrics, including ImageNet accuracy and AGE metrics. We strictly follow (Yi et al., 2024; Zohar et al., 2023) to perform Leave-One-Out evaluation on the 23 datasets, where the coefficients of metrics on the evaluated dataset is decided by regression on the rest. Inspired by (Yi et al., 2024), we adopt Huber and ridge regression to fit the coefficients of different metrics. For a fair comparison, we adopt the same prompt templates to construct text classifiers as in (Yi et al., 2024; Zohar et al., 2023). More details are provided in the appendix.

**Result analysis.** Table 1 demonstrates performance comparison between SAGE and baselines. Notably, SAGE performs better than the three baselines across all criteria without the need for LLM to generate texts. Note that all methods incorporate ImageNet information during model selection since ModelGPT and SWAB have combined ImageNet accuracy. Therefore, SAGE does not leverage extra information by utilizing ImageNet images as proxies of target images. We also combine SAGE and ModelGPT by combining AGE metrics with the text scores used in ModelGPT, which significantly boosts VLM selection performance. As shown in Table 1, the AGE metrics and text scores enjoy complementary advantages.

### 5.2 Ablation Studies

In this section, we provide ablation experiment results of the designs of SAGE to justify our choices. We present the benchmark criteria for SAGE and SAGE$^+$ (representing combining AGE scores and the text scores derived from LLM-generated texts in (Zohar et al., 2023)).

#### 5.2.1 Main Technical Designs in SAGE

In the methodology section, we introduce our core mechanism **AGE** which quantifies similarity between individual VLMs and their ensemble counterpart on in-the-wild images and target task descriptions. In addition, to improve AGE as a proxy metric for VLM selection, we introduce several technical designs. The main technical designs in SAGE are as follows:

- **Semantic Retrieval (SR)**: We prioritize images from $\mathcal{X}'$ semantically closer to the task via weighting, using category names $\mathcal{C}$.

- **Composed Similarity (CS)**: We combine extra similarity measures in addition to Top-1 accuracy for robust estimation of AGE.

- **Regularized Regression (RR)**: Inspired by (Yi et al., 2024), we use Ridge regression and Huber regression instead of vanilla linear regression to combine multiple metrics.

Table 2: Ablation results of the main designs

| SR | CS | RR | SAGE | | SAGE$^+$ | |
|---|---|---|---|---|---|---|
| | | | $R_5$ | $\tau$ | $R_5$ | $\tau$ |
| $\times$ | $\times$ | $\times$ | 53.9 | 25.0 | 53.9 | 28.1 |
| $\times$ | $\times$ | $\checkmark$ | 53.9 | 25.0 | 53.9 | 34.8 |
| $\times$ | $\checkmark$ | $\times$ | 53.9 | 29.6 | 54.8 | 34.8 |
| $\times$ | $\checkmark$ | $\checkmark$ | 53.9 | 31.0 | 56.5 | 39.4 |
| $\checkmark$ | $\times$ | $\times$ | 53.0 | 24.6 | 53.9 | 27.5 |
| $\checkmark$ | $\times$ | $\checkmark$ | 53.0 | 27.5 | 53.9 | 34.8 |
| $\checkmark$ | $\checkmark$ | $\times$ | **54.8** | 31.0 | 54.8 | 35.7 |
| $\checkmark$ | $\checkmark$ | $\checkmark$ | **54.8** | **31.6** | **57.4** | **41.4** |

Table 3: Ablation results of the AGE scores

| CC | KL | TV | SAGE | | SAGE$^+$ | |
|---|---|---|---|---|---|---|
| | | | $R_5$ | $\tau$ | $R_5$ | $\tau$ |
| $\times$ | $\times$ | $\times$ | 53.0 | 27.5 | 53.9 | 34.8 |
| $\times$ | $\times$ | $\checkmark$ | 53.0 | 27.5 | 53.9 | 35.0 |
| $\times$ | $\checkmark$ | $\times$ | 53.0 | 27.5 | 53.9 | 38.5 |
| $\times$ | $\checkmark$ | $\checkmark$ | 53.0 | 27.5 | 53.9 | 39.4 |
| $\checkmark$ | $\times$ | $\times$ | 53.0 | 31.0 | 56.5 | 35.9 |
| $\checkmark$ | $\times$ | $\checkmark$ | **54.8** | 31.0 | 56.5 | 36.5 |
| $\checkmark$ | $\checkmark$ | $\times$ | 53.0 | 31.0 | **57.4** | 39.4 |
| $\checkmark$ | $\checkmark$ | $\checkmark$ | **54.8** | **31.6** | **57.4** | **41.4** |

Table 4: Comparison of methods under the small model pool setting

| Method | $R_5$ | $\tau$ | $R_5 + \tau$ |
|---|---|---|---|
| ImageNet accuracy | 53.9 | 30.5 | 84.4 |
| ModelGPT | 56.5 | 34.8 | 91.3 |
| SAGE | 60.0 | **50.9** | 110.9 |
| SAGE +ModelGPT | **63.5** | **50.9** | **114.4** |

We present the ablation results in Table 2, which confirm that these designs are essential for SAGE.

### 5.2.2 SIMILARITY SCORES IN SAGE

In the methodology section, we propose using multiple metrics to quantify the similarity between the predictions of individual and ensemble models to enhance robustness. We introduce the following three extended similarity scores to complement vanilla AGE:

- **Class Ranking Correlation (CC)**: The proportion of mutual items in the Top-$k$ classes predicted by the individual and the ensemble. We set $k \in \{2, 3\}$ in SAGE in addition to $k = 1$ (vanilla AGE).

- **Exponential of Negative KL Divergence (KL)**: Similarity induced by KL-divergence.

- **Normalized Total Variance Distance (TV)**: Similarity induced by total variance distance.

We present ablation results in Table 3, which demonstrate that these similarity scores are complementary in SAGE. In the base case (without applying similarity scores), we combine ImageNet accuracy with vanilla AGE (induced by pseudo-accuracy) for SAGE, and further incorporate text scores for SAGE$^+$.

### 5.3 SIGNIFICANCE EXPERIMENTS

#### 5.3.1 RESULTS ON SMALLER MODEL POOLS

Although SAGE involves VLM ensembles, it is insensitive to the construction of the VLM zoo. To demonstrate this, we select a subset from the full group of 35 models, excluding the "strong models" to form a "small model pool." As shown in Table 4, SAGE still significantly outperforms the baselines under this setting. The construction of the subset is described in the appendix.

#### 5.3.2 STABILITY ON THE CHOICE OF IN-THE-WILD IMAGES

The set of in-the-wild images used for SAGE is randomly sampled from ImageNet1K, with one image per class. In Table 5, we conduct 20 random experiments using different samples to evaluate the stability of SAGE with respect to the randomness of the in-the-wild data. Results indicate that SAGE is stable with regard to the choice of in-the-wild images.

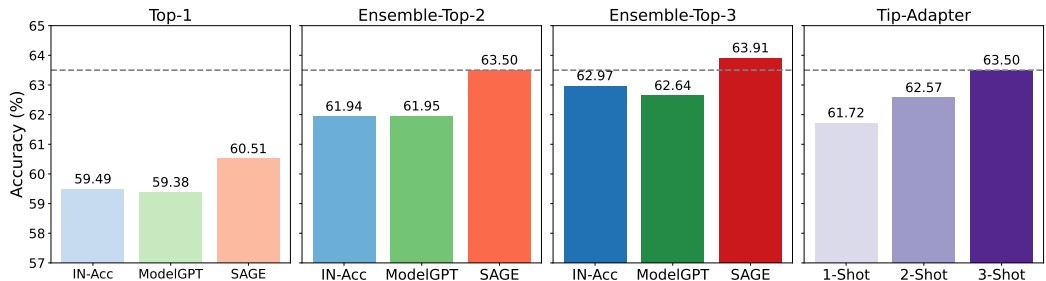

Figure 4: **Accuracy of different methods.** The y-axis represents average accuracy across 23 datasets. The left three figures are: accuracy of Top-1 model, Top-2 ensembles, and Top-3 ensembles selected by ImageNet accuracy, ModelGPT, and SAGE. We additionally evaluate Tip-Adapter as a few-shot adaptation baseline for reference.

Table 5: Stability of SAGE under randomness of in-the-wild images

| Method | $R_5$ | $\tau$ | $R_5 + \tau$ |
|---|---|---|---|
| SAGE | $0.548 \pm 0.007$ | $0.316 \pm 0.016$ | $0.829 \pm 0.02$ |
| SAGE+ModelGPT | $0.563 \pm 0.006$ | $0.365 \pm 0.017$ | $0.928 \pm 0.022$ |

## 5.4 FROM ENSEMBLES FOR SELECTION TO SELECTION FOR ENSEMBLES

We have shown that ensembles can benefit VLM selection, and in practice, ensemble methods are widely used to improve VLM performance (Lu et al., 2024; Jeong et al., 2024; Li et al., 2023; Huang et al., 2022b). Here, we show that model rankings from SAGE can be used to construct superior VLM ensembles compared to baselines.

**Building ensembles from model ranks.** In selective ensemble methods (Caruana et al., 2004; Wood et al., 2023), ranking-based approaches aim to generate a model ranking such that selecting the Top-$k$ models yields a strong ensemble.

**SAGE produces better ensembles.** We construct Top-$k$ ensembles using rankings from ImageNet accuracy, ModelGPT (Zohar et al., 2023), and SAGE, following (Lu et al., 2024) with zero-shot output averaging. Evaluated on 23 datasets, Figure 4 shows that SAGE consistently produces superior Top-$k$ ensembles.

**Comparison to few-shot adaptation.** Few-shot adaptation methods (Gao et al., 2024; Zhang et al., 2022; Silva-Rodriguez et al., 2024) fine-tune VLMs with limited target samples. Using Tip-Adapter (Zhang et al., 2022) on the Top-1 model by ImageNet accuracy as a baseline, we report averages over 30 trials. As shown in Figure 4, ensembles from SAGE achieve performance comparable to Tip-Adapter with 3-shot samples, demonstrating that zero-shot ensembling can rival few-shot adaptation and compensate for data scarcity by increasing the number of models.

## 6 CONCLUSION

We address the challenge of *zero-shot* VLM selection for image recognition tasks. We propose Selection via AGreement-with-the-Ensemble (SAGE), a method that quantifies the similarity between individual VLMs and their ensemble counterparts using in-the-wild images and target category names. We provide empirical evidence to support our method. SAGE outperforms existing approaches on established benchmarks.

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

## A  APPENDIX

### A.1  USE OF LLMS

LLMs are used to assist in polishing writing, including grammar checking and improving expression.

### A.2  MODELS AND DATASETS

Following Zohar et al. (2023), we use the OpenCLIP library Ilharco et al. (2021) to form a VLM zoo. As shown in Table 6, these VLMs differ in terms of model architecture (e.g., ResNet He et al. (2016b), Transformer Vaswani et al. (2017), ConvNext Liu et al. (2022)), pretraining datasets (e.g., OpenAI's data Radford et al. (2021), LAION 2B Schuhmann et al. (2021)), and other factors. As shown in Table 7, LOVM utilizes 23 datasets from different domains. Table 6 and Table 7 are from Zohar et al. (2023) and Yi et al. (2024).

### A.3  IMPLEMENTATION DETAILS OF SAGE

**Semantic retrieval.** We calculate the semantic similarity between samples in the ImageNet subset and the textual prompts $\mathcal{T}$ of target tasks using a VLM $f_0$ as:

$$w(\mathbf{x}) = \frac{\exp\left(\sum_{\mathbf{t} \in \mathcal{T}} \text{sim}(\mathbf{x}, \mathbf{t}; f_0)/C\right)}{\sum_{\mathbf{x} \sim \mathcal{X}'_{\text{IN}}} \exp\left(\sum_{\mathbf{t} \in \mathcal{T}} \text{sim}(\mathbf{x}, \mathbf{t}; f_0)/C\right)} . \tag{12}$$

Table 6: The 35 VLMs used in LOVM.

| ID | Model | Name | Dataset | Name |
|----|-------|------|---------|------|
| 1 | RN50 | RN50 | openai | WIT |
| 2 | RN101 | RN101 | openai | WIT |
| 3 | RN50x4 | RN50x4 | openai | WIT |
| 4 | RN50-16 | RN50x16 | openai | WIT |
| 5 | RN50x64 | RN50x64 | openai | WIT |
| 6 | ViT-B-32 | ViT-B/32 | laion400m_e31 | L400m |
| 7 | ViT-B-32 | ViT-B/32 | laion400m_e32 | L400m |
| 8 | ViT-B-32-quickgelu | ViT-B/32 | laion400m_e32 | L400m |
| 9 | ViT-B-32 | ViT-B/32 | openai | WIT |
| 10 | ViT-B-32 | ViT-B/32 | laion2b_s34b_b79k | L2b-b |
| 11 | ViT-B-32 | ViT-B/32 | laion2b_e16 | L2b-c |
| 12 | ViT-B-16 | ViT-B/16 | laion400m_e32 | L400m |
| 13 | ViT-B-16 | ViT-B/16 | openai | WIT |
| 14 | ViT-B-16-240 | ViT-B/16-240 | laion400m_e32 | L400m |
| 15 | ViT-L-14 | ViT-L/14 | laion400m_e31 | L400m |
| 16 | ViT-L-14 | ViT-L/14 | laion400m_e32 | L400m |
| 17 | ViT-L-14 | ViT-L/14 | laion2b_s32b_b82k | L2b-b |
| 18 | ViT-L-14 | ViT-L/14 | openai | WIT |
| 19 | ViT-L-14-336 | ViT-L/14-336 | openai | WIT |
| 20 | ViT-G-14 | ViT-G/14 | laion2b_s12b_b42k | L2b-a |
| 21 | ViT-G-14 | ViT-G/14 | laion2b_s34b_b88k | L2b-a |
| 22 | ViT-H-14 | ViT-H/14 | laion2b_s32b_b79k | L2b-b |
| 23 | coca_ViT-B-32 | CoCa-ViT-B/32 | laion2b_s13b_b90k | L2b-c |
| 24 | coca_ViT-B-32 | CoCa-ViT-B/32 | mscoco_finetuned_laion2b_s13b_b90k | L2b-c + coco |
| 25 | coca_ViT-L-14 | CoCa-ViT-L/14 | laion2b_s13b_b90k | L2b-c |
| 26 | coca_ViT-L-14 | CoCa-ViT-L/14 | mscoco_finetuned_laion2b_s13b_b90k | L2b-c + coco |
| 27 | convnext_base | ConvNEXT-B | laion400m_s13b_b51k | L400m-c |
| 28 | convnext_base_w | ConvNEXT-BW | laion2b_s13b_b82k | L2b-d |
| 29 | convnext_base_w | ConvNEXT-BW | laion2b_s13b_b82k_augreg | L2b-e |
| 30 | convnext_base_w | ConvNEXT-BW | laion_aesthetic_s13b_b82k | L2b-f |
| 31 | convnext_base_w_320 | ConvNEXT-BW-320 | laion_aesthetic_s13b_b82k | L2b-f |
| 32 | convnext_base_w_320 | ConvNEXT-BW-320 | laion_aesthetic_s13b_b82k_augreg | L2b-g |
| 33 | convnext_large_d | ConvNEXT-LD | laion2b_s26b_b102k_augreg | L2b-h |
| 34 | convnext_large_d_320 | ConvNEXT-LD-320 | laion2b_s29b_b131k_ft | L2b-i |
| 35 | convnext_large_d_320 | ConvNEXT-LD-320 | laion2b_s29b_b131k_ft_soup | L2b-j |

Here, $f_0$ can be any VLM in the model zoo since the image embeddings are precomputed and we infer $\mathcal{T}$ through all VLMs during VLM selection. We recommend to use large VLMs to ensure the quality of cross-modal embedding. In terms of implementation in this paper, we choose ViT-H/14-L2b-b (ID 22 in Table 6) as $f_0$.

**Regression methods.** To combine multiple proxy metrics in the Leave-One-Out evaluation, we follow Yi et al. (2024) and apply regularized regression methods. We apply Ridge regression with $\alpha = 1e - 5$ for SAGE to combine ImageNet accuracy with AGE metrics. We apply Huber regression with $\alpha = 1.15$ for SAGE+ModelGPT as in Yi et al. (2024), which combines ImageNet accuracy, AGE metrics and text scores from ModelGPT.

### A.4 CONSTRUCTION OF THE SMALL MODEL POOL

The small model pool described in Section 5.3 consists of all models in Table 6 except for the following (by ID): 5, 18, 19, 20, 21, 22, 31, 32, 33, 34, and 35. These excluded models exhibit significantly higher average performance across the datasets.

### A.5 DISCUSSIONS ON THE METHODOLOGY

**Why not use labels from the original ImageNet dataset?** In SAGE, we utilize ImageNet images as in-the-wild images and discard their labels in the original ImageNet dataset. We argue that there is a gap between the ImageNet classes $\mathcal{T}_{\text{IN}}$ and the target task $\mathcal{T}$. For example, in a car brand classification task, an ImageNet image labeled as "car" may be useful for evaluating VLMs. However, the label "car"

Table 7: The 23 tasks used in LOVM.

| Dataset | Classes | Task | Domain |
|---|---|---|---|
| Imagenet (Deng et al., 2009) | 1000 | classification | natural image |
| SUN397 (Xiao et al., 2010) | 397 | scene und. | natural image |
| Country211 (Radford et al., 2021) | 211 | geolocation | natural image |
| Stanford Cars (Krause et al., 2013) | 196 | classification | natural image |
| Flowers102 (Nilsback & Zisserman, 2008) | 102 | classification | natural image |
| CIFAR100 (Krizhevsky, 2009) | 100 | classification | natural image |
| DTD (Cimpoi et al., 2014) | 46 | classification | textural image |
| RESISC45 (Cheng et al., 2017) | 45 | classification | satellite images |
| GTSRB (Stallkamp et al., 2011) | 43 | classification | natural image |
| Oxford Pets (Parkhi et al., 2012) | 37 | classification | natural image |
| VOC2007 (Everingham et al., 2007) | 20 | classification | natural image |
| STL10 (Coates et al., 2011) | 10 | classification | natural image |
| EuroSAT (Helber et al., 2019) | 10 | classification | satellite images |
| MNIST (LeCun et al., 2010) | 10 | classification | hand-writing |
| SVHN (Netzer et al., 2011) | 10 | OCR | natural image |
| CLEVR-C (Johnson et al., 2017) | 8 | object counting | natural image |
| CLEVR-D (Johnson et al., 2017) | 8 | distance est. | natural image |
| FER2013 (Goodfellow et al., 2013) | 7 | fac. exp. rec. | natural image |
| DMLab (Zhai et al., 2020) | 6 | distance est. | synthetic |
| Retinopathy (Kaggle & EyePacs, 2015) | 5 | classification | retina scan |
| KITTI (Geiger et al., 2013) | 4 | distance est. | natural image |
| PCam (Veeling et al., 2018) | 2 | classification | histopathology |
| Rendered SST2 (Radford et al., 2021) | 2 | OCR | text image |

does not provide enough detail to determine the specific brand. Furthermore, valuable information in a natural image is not always reflected in its original label. For instance, a photo of a car parked in the mud (labeled "car") could be helpful for a land-use classification task, even though the label provides no clue about the context.

**Inference cost of SAGE compared to previous methods.** LLM-based methods like ModelGPT Zohar et al. (2023) mainly involve the following procedures. First, they need to generate a large-scale caption dataset ($K$-shot for each of the $C$ classes) with an LLM. Then, they need to feed the $K \times C$ texts along with the text descriptions $\mathcal{T}$ to the text encoders of the $M$ VLMs in the model zoo. In contrast, SAGE does not require access to an LLM interface. Additionally, the embeddings of the in-the-wild images $\mathcal{X}'$ remain invariant across different downstream tasks and can therefore be precomputed offline. As a result, the only online inference cost in SAGE is computing the embeddings of the task descriptions $\mathcal{T}$. Thus, SAGE does not increase the overall cost compared to previous LLM-based methods.

A.6 EXPERIMENTS COMPUTE RESOURCES

We conduct our experiments on a single NVIDIA RTX 4090 GPU (24GB). The primary computational cost arises from performing inference on the ImageNet-1K sample images across 35 VLMs, which takes approximately 3 GPU hours. The remaining components of our experiments are lightweight and can be executed with minimal computational overhead.

