# OpenReview forum: "Agreement with the Ensemble for Zero-Shot Vision-Language Model Selection"
_ICLR.cc/2026/Conference — Submitted to ICLR 2026_

### Official Review · Reviewer_gTpr · 2025-10-25

**Soundness:** 2
**Presentation:** 3
**Contribution:** 2
**Rating:** 4
**Confidence:** 5

**Summary:**

This paper proposes SAGE (Selection via Agreement-with-the-Ensemble), a method for zero-shot vision–language model (VLM) selection. Instead of relying on text-only proxies (e.g., LLM-generated captions), SAGE computes an Agreement-with-Ensemble (AGE) metric using in-the-wild images and measures how consistent each model’s predictions are with those of a model ensemble. Experiments on 35 VLMs across 23 datasets show improved correlation with true accuracy and outperform existing baselines (ModelGPT, SWAB).

**Strengths:**

1. Addresses a practically relevant and underexplored problem: zero-shot VLM selection without target images or labels.
2. The AGE metric is simple, intuitive, and easy to compute without LLMs.
3. Extensive experiments with consistent improvements over baselines.

**Weaknesses:**

1. The effectiveness of the SAGE method largely relies on a key assumption: that the ensemble of vision–language models can provide high quality pseudo labels for in-the-wild images. However, the reliability of such ensemble generated pseudo labels is not always guaranteed.
2. Using ImageNet images (even unlabeled) introduces domain leakage: most target datasets are natural images, making results less convincing.
3. The idea of using agreement or disagreement as a proxy for generalization has strong precedents—e.g., “Assessing Generalization of SGD via Disagreement” (Jiang et al., ICLR 2022) and AETTA (Li et al., 2024), yet these are not cited or compared.
4. Only empirical correlations are shown, no formal analysis of why agreement implies accuracy.

**Questions:**

1. How does SAGE perform when the in-the-wild images come from a very different domain (e.g., satellite, medical)?
2. Would the performance drop significantly if ImageNet were replaced with a more diverse or less curated dataset such as LAION or COCO?
3. Have you evaluated whether including the tested model in the ensemble biases the AGE score?

---

> ### Author Response · Authors · 2025-11-21
>
> **Dear Reviewer gTpr**:
>
> Thank you for the valuable feedback on our paper. We appreciate the time and effort you have put into reviewing our work. We have carefully read your review and addressed your concerns as follows.
>
> ---
>
> **W1**: We agree that SAGE relies on the quality of pseudo labels generated by the model ensemble. Our empirical study shows that this assumption is sufficiently **robust** in practice: SAGE consistently outperforms existing baselines across diverse datasets, and the significance tests reported in the paper further validate the reliability of ensemble-generated signals.
>
> ---
>
> **W2**: The standard benchmark in Section 5.1 includes 11 non-natural datasets (Table 7 in the appendix). We report their aggregated results below:
>
> | Metric | INB | ModelGPT | SAGE | SAGE+ModelGPT |
> |--------|------|-----------|--------|----------------|
> | **R**        | 30.0 | 32.9 | 37.1 | **40.0** |
> | **τ**        | 9.5 | 14.3 | 16.7 | **28.6** |
> | **R+τ**      | 39.5 | 47.1 | 53.8 | **68.6** |
>
> (INB refers to the ImageNet Accuracy baseline.)
>
> Although these datasets are challenging for all methods, SAGE and SAGE+ModelGPT still outperform existing baselines. We will highlight this result more clearly.
>
> ---
>
> **W3**: Thank you for pointing out the related literature. Prior work (e.g., Jiang et al., 2022; Li et al., 2024) indeed investigates (dis)agreement as a proxy for generalization, but all such methods require access to **target test images** (albeit unlabeled). In contrast, our setting is fundamentally different: SAGE performs model selection **without** using any target-domain images. This distinction is crucial, as existing agreement-based approaches are not directly applicable in the zero-shot setting, and it highlights an important contribution of our work. We will cite and discuss these works explicitly in the final version.
>
>
> ---
>
> **W4**: We appreciate the suggestion. Our method is primarily motivated by a straightforward intuition — models that behave similarly to a strong ensemble tend to generalize well — and validated through extensive empirical evaluations showing significant gains over state-of-the-art baselines. While empirical contributions align with ICLR’s scope, we agree that theoretical insights would further strengthen our work. We have a preliminary theoretical analysis and will include it in the appendix in the next revision.
>
> ---
>
> **Q1**: Our ablation shows that semantic retrieval effectively reduces the domain gap between in-the-wild images and downstream tasks. However, if the in-the-wild data comes from a highly specialized domain (e.g., medical, satellite), this gap becomes difficult to bridge, limiting adaptability and degrading performance. As discussed in Section 4.2, a diverse set of in-the-wild images is essential for broad generalization.
>
>
> ---
>
> **Q2**: We report results using LAION-10K and COCO-minitrain (25k) as in-the-wild images:
>
> | LAION | SAGE | SAGE+ModelGPT |
> |--------|-------|----------------|
> | **R**      | 50.4 | **54.8** |
> | **τ**      | 31.0 | **36.5** |
> | **R+τ**    | 81.4 | **91.3** |
>
>
>
> | COCO | SAGE | SAGE+ModelGPT |
> |--------|-------|----------------|
> | **R**      | 53.9 | **54.8** |
> | **τ**      | 29.9 | **36.8** |
> | **R+τ**    | 83.8 | **91.6** |
>
> ---
>
> **Q3**: Our default design includes the tested model in the ensemble for fairness and simplicity. We also experimented with excluding the tested model and observed no significant performance difference. Because the ensemble includes a large number of models, removing one model has negligible effect. We will clarify this design choice in the paper.

---

> > ### Comment · Reviewer_gTpr · 2025-11-27
> >
> > Thank you for the rebuttal. I appreciate the additional results and clarifications. However, my main concern remains that the core motivation of the method, namely relying on ensemble generated pseudo labels as a reliable signal for zero shot model selection, is not fully justified by the provided analysis. This foundational assumption still appears insufficiently supported.
> >
> > Given this, my overall evaluation and score remain unchanged.

---

### Official Review · Reviewer_3tTy · 2025-10-26

**Soundness:** 3
**Presentation:** 3
**Contribution:** 3
**Rating:** 6
**Confidence:** 3

**Summary:**

The paper tackles the problem of zero-shot VLM selection, where the goal is to choose the most suitable pretrained VLM for a downstream classification task without access to target images or labels, using only the category names.Existing approaches such as ModelGPT and SWAB rely on text-only proxy evaluations using LLM-generated captions, but these methods suffer from the modality gap between text and image domains.
To address this, the authors propose SAGE (Selection via AGreement-with-the-Ensemble), which introduces the AGE metric which is a measure of how much an individual VLM’s prediction agrees with the predictions of an ensemble of VLMs. The key insight is that ensemble consensus provides high-quality pseudo-labels that correlate strongly with actual performance.
SAGE extends AGE to the zero-shot setting by:
1) Using in-the-wild images (e.g., ImageNet samples) to approximate unseen target data
2) Employing semantic retrieval to select or weight images semantically relevant to the task categories

Incorporating robust similarity measures (class ranking correlation, KL divergence, total variation) to improve stability
The paper conducts experiments across 23 benchmark datasets and a zoo of 35 VLMs which show that SAGE outperforms baselines like ModelGPT and SWAB.

**Strengths:**

1) Introduces a novel ensemble-agreement metric (AGE) for model evaluation without labels. It moves away from reliance on LLMs, reducing computational cost and potential biases.
2) The paper has Conducts thorough ablations showing the importance of each component: semantic retrieval, composed similarity, and regularized regression.
3) The method demonstrates robustness to random image sampling and insensitivity to model pool composition.

**Weaknesses:**

1) In Eq 7, the semantic retrieval weighting depends on a reference VLM f_0. The choice of f_0 may inference results, but this sensitivity is not analyzed.
2) Regarding the usage of “in-the-wild” ImageNet images, their domain proximity to training data may bias the results. How does the SAGE performance vary on natural images and non-natural image domains (e.g. satellite, medical, sketch, etc.)?
3) The ensemble-based metric may implicitly leak information about overall VLM capacity. The paper could benefit from analysis on whether SAGE generalizes to unseen architectures or novel domains.

**Questions:**

1) The pseudo-label assumption assumes that ensemble consensus correlates with accuracy. Could this break down if all models share the same bias? Since AGE relies on ensemble agreement, could a biased ensemble (e.g., models trained on similar data) lead to misleading pseudo-labels?
2) For the combination with text-based scores (SAGE + ModelGPT), how much of the improvement comes from complementarity vs. redundancy?

---

> ### Author Response · Authors · 2025-11-21
>
> **Dear Reviewer 3tTy**:
>
> Thank you for the valuable feedback on our paper. We appreciate the time and effort you have put into reviewing our work. We have carefully read your review and addressed your concerns as follows.
>
> ---
>
> **W1**. Thank you for raising this point. We provide an additional analysis using several averagely well-performing models in the zoo (as recommended in Appendix Table 6) as alternative \( f_0 \). Results are shown below:
>
> | ID | SAGE-R | SAGE-τ | SAGE+ModelGPT-R | SAGE+ModelGPT-τ |
> |----|--------|--------|------------------|------------------|
> | 19 | 53.0   | 31.0   | 56.5             | 35.9             |
> | 20 | 53.9   | 31.0   | 56.5             | 36.5             |
> | 21 | 53.9   | 31.0   | 56.5             | 39.8             |
> | 22 | 54.8   | 31.6   | 57.4             | 41.4             |
>
> The performance is stable across different choices of \( f_0 \), suggesting SAGE is **not highly sensitive** to this selection. We will include this robustness analysis.
>
> ---
>
> **W2**. We agree that model selection on non-natural images is more challenging for all methods. Below are the aggregated results over 11 non-natural datasets on the standard benchmark (see Table 7 in the appendix):
>
> | Metric | INB | ModelGPT | SAGE | SAGE+ModelGPT |
> |--------|------|-----------|--------|----------------|
> | **R**        | 30.0 | 32.9 | 37.1 | **40.0** |
> | **τ**        | 9.5  | 14.3 | 16.7 | **28.6** |
> | **R+τ**      | 39.5 | 47.1 | 53.8 | **68.6** |
>
> (INB refers to the ImageNet Accuracy baseline.)
>
> SAGE and SAGE+ModelGPT achieve the best performance even in these difficult settings. We will highlight this comparison more clearly.
>
> ---
>
> **W3**. We appreciate the reviewer’s suggestion.
>
> - **Novel domains:** As shown in W2, SAGE outperforms baselines on non-natural datasets. We plan additional experiments on more diverse domains in future revisions.
> - **Unseen architectures:** SAGE only requires that a VLM provide (i) image–text similarity scores or (ii) class-probability predictions. Thus the method is naturally applicable beyond CLIP-like models. We plan additional experiments on more architectures in future revisions.
>
> ---
>
>
> **Q1**. This is possible in theory; however, in our study, the pretrained CLIP zoo is highly **diverse** thanks to the community (as described in *Introduction*). We empirically observe that within a diverse model zoo, AGE reliably correlates with accuracy.
>
> ---
>
> **Q2**. Two observations support complementarity:
>
> 1. **SAGE consistently outperforms ModelGPT**, indicating it captures unique signals.
> 2. **SAGE+ModelGPT substantially improves over SAGE**, suggesting the text-based scores contribute additional, non-redundant information.
>
> These results indicate that SAGE and ModelGPT capture **different and complementary aspects** of model selection quality.

---

### Official Review · Reviewer_uSAU · 2025-10-31

**Soundness:** 3
**Presentation:** 3
**Contribution:** 2
**Rating:** 4
**Confidence:** 3

**Summary:**

This paper proposes SAGE, a method to select top-performing vision-language models (VLMs) for zero-shot downstream inference from a VLM model zoo. The proposed method leverages in-the-wild images with pseudo-labels generated by VLM ensembles to estimate the performance for each individual VLM. SAGE further conducts semantic retrieval to reduce the semantic gap between the in-the-wild image dataset and the target task. Compared with state-of-the-art approaches, SAGE achieves higher Recall 5 and Weighted Kendall's coefficient on 23 downstream datasets.

**Strengths:**

- The problem being investigated, i.e., how to select proper models from a model zoo, is becoming more critical at this time. Research towards this direction should be encouraged.
- The paper is well-written and easy to follow, and the idea is simple yet effective on the evaluated 23 downstream datasets.

**Weaknesses:**

- The effectiveness of the proposed method relies heavily on the constructed in-the-wild image dataset. The weight estimation in eq. (7) could become unreliable if the semantic gap between the in-the-wild dataset (e.g., ImageNet) and the target task (e.g., medical images) is huge. To improve robustness, the authors should consider developing a principled method to construct the proxy dataset. For instance, by curating web images relevant to the target categories, rather than using a random subset of ImageNet.
- The result in table 4 is insufficient to demonstrate that the VLM ensemble used in SAGE is insensitive to the construction of the model zoo. The relationship between model selection performance and the size of the ensemble should be discussed.

**Questions:**

- Will the selected model still performs consistently better than other models in the zoo after few-shot finetuning?

---

> ### Author Response · Authors · 2025-11-21
>
> **Dear Reviewer uSAU**:
>
> Thank you for the valuable feedback on our paper. We appreciate the time and effort you have put into reviewing our work. We have carefully read your review and addressed your concerns as follows.
>
> ---
>
> **W1**: We appreciate the reviewer’s considerations regarding the dependence on the in-the-wild dataset. Our main argument is that SAGE remains **robust** as long as the in-the-wild dataset is comprehensive, which our experiments empirically support  (See Table 5). That said, we agree that incorporating a broader and more diverse collection of images — or adopting a more principled dataset-construction strategy — may further enhance performance, especially for domains with large semantic gaps. Exploring domain-aware or web-curated proxy datasets is a promising direction, and we consider this an important avenue for future work.
>
> ---
>
> **W2**: We thank the reviewer for pointing out the need to more clearly characterize the sensitivity of SAGE to the model-zoo size. In response, we provide additional experiments where we create multiple random subsets of varying sizes, repeat the evaluation, and report averaged results (shown below).
>
> We observe that when the model-zoo size is small (<20), SAGE and the baselines perform similarly. However, as the number of available models increases, the advantage of SAGE becomes increasingly significant, consistently outperforming INB and ModelGPT across metrics.
>
> | size | INB-R | ModelGPT-R | SAGE-R | INB-tau | ModelGPT-tau | SAGE-tau | INB-sum | ModelGPT-sum | SAGE-sum |
> |------|--------|-------------|------------|---------|--------------|----------|---------|--------------|----------|
> | 10 | 73.9 | 75.7 | 74.5 | 32.7 | 40.0 | 41.0 | 106.6 | 114.5 | **116.7** |
> | 11 | 72.8 | 74.2 | 73.3 | 34.2 | 39.9 | 43.1 | 106.9 | 113.2 | **117.4** |
> | 12 | 69.6 | 70.7 | 70.1 | 32.1 | 39.3 | 40.2 | 101.7 | 109.4 | **110.9** |
> | 13 | 67.5 | 69.6 | 67.8 | 30.9 | 38.3 | 38.7 | 98.5 | 106.1 | **108.3** |
> | 14 | 63.8 | 65.5 | 64.6 | 29.6 | 33.4 | 31.1 | 93.4 | **98.0** | 96.6 |
> | 15 | 59.1 | 61.7 | 60.9 | 28.2 | 34.7 | 34.6 | 87.3 | 95.6 | **96.3** |
> | 16 | 57.9 | 59.7 | 59.7 | 30.2 | 33.4 | 31.1 | 88.2 | **93.1** | 90.8 |
> | 17 | 58.6 | 60.3 | 59.4 | 27.2 | 29.7 | 27.9 | 85.8 | **89.1** | 88.2 |
> | 18 | 59.1 | 60.6 | 60.0 | 25.5 | 29.6 | 26.7 | 84.6 | **89.6** | 87.2 |
> | 19 | 58.3 | 60.6 | 59.1 | 21.9 | 26.8 | 25.4 | 80.2 | 85.9 | **86.0** |
> | 20 | 58.3 | 59.4 | 59.1 | 25.6 | 29.3 | 32.1 | 83.9 | 88.4 | **91.5** |
> | 21 | 57.4 | 58.6 | 57.9 | 23.3 | 26.0 | 27.8 | 80.7 | 83.9 | **86.4** |
> | 22 | 56.8 | 59.1 | 57.7 | 21.7 | 26.5 | 28.8 | 78.6 | 84.2 | **87.9** |
> | 23 | 55.9 | 58.6 | 56.8 | 22.7 | 26.6 | 31.4 | 78.6 | 83.4 | **89.9** |
> | 24 | 55.1 | 57.9 | 56.2 | 21.7 | 27.5 | 30.2 | 76.8 | 83.8 | **88.2** |
> | 25 | 55.1 | 57.7 | 55.9 | 21.7 | 25.8 | 29.4 | 76.8 | 81.7 | **87.1** |
> | 26 | 55.1 | 57.7 | 55.9 | 21.7 | 26.2 | 29.1 | 76.8 | 82.1 | **86.8** |
> | 27 | 53.9 | 56.8 | 55.1 | 23.4 | 27.1 | 30.2 | 77.3 | 82.1 | **87.1** |
> | 28 | 52.2 | 55.1 | 53.9 | 22.7 | 22.4 | 30.4 | 74.9 | 76.3 | **85.5** |
> | 29 | 52.5 | 55.7 | 54.2 | 25.2 | 25.6 | 33.2 | 77.7 | 79.8 | **88.9** |
> | 30 | 51.3 | 55.1 | 53.6 | 26.1 | 26.0 | 34.1 | 77.4 | 79.6 | **89.2** |
> | 31 | 50.4 | 54.5 | 53.0 | 24.2 | 25.8 | 31.4 | 74.6 | 78.8 | **85.9** |
> | 32 | 50.1 | 54.2 | 52.8 | 24.2 | 25.2 | 31.6 | 74.3 | 77.9 | **86.0** |
> | 33 | 50.1 | 54.5 | 52.2 | 24.2 | 24.7 | 31.5 | 74.3 | 76.9 | **86.0** |
> | 34 | 49.3 | 53.6 | 51.0 | 24.6 | 24.6 | 32.1 | 73.9 | 75.7 | **85.7** |
>
>
> (INB refers to the ImageNet Accuracy baseline.)
>
> ---
>
> **Q1**: We agree that the ability to predict few-shot performance from pure zero-shot information would be highly valuable. However, once target data are available for tuning, it is natural to consider using the same data for model selection as well. Although this is not the primary focus of our work—which deliberately avoids using target images—we provide additional results for completeness.
>
> Specifically, we keep the same zero-shot selection procedure but replace the ground-truth ranking with the ranking obtained after 2-shot tuning using Tip-Adapter.
>
> | Metric | INB  | ModelGPT | SAGE | SAGE+ModelGPT |
> |--------|------|-----------|------|----------------|
> | R      | 46.1 | 50.4      | 52.2 | **53.9**       |
> | τ      | 20.3 | 33.9      | 32.5 | **38.8**       |
> | R+τ    | 66.4 | 84.3      | 84.6 | **92.8**       |

---

### Comment · Area_Chair_z389 · 2025-11-25
**Encourage discussions**

Hi all,

The authors have submitted their responses. Please take a moment to review them and see if they address your concerns.

Your thoughtful input is essential for a successful reviewing process and is greatly appreciated.

Many thanks,

Area Chair

---

### Meta-Review · Area_Chair_a37K · 2026-01-03

**Summary:**

This paper received scores of 4,6,4.  Initial concerns include dependence on the in-the-wild proxy dataset, sensitivity to model-zoo construction, sensitivity to the reference VLM, bias from ImageNet “in-the-wild” images, generalization to unseen architectures / domains, reliability of ensemble pseudo-labels, missing related work, and lack of theory.

**Reviewer Concerns:**

Many concerns were addressed by the rebuttal.  However, a fundamental concern remains raised by multiple reviewers: the work relies heavily on the assumption that ensemble-generated pseudo labels are reliable for zero-shot model selection, but this assumption is not convincingly supported by the paper / rebuttal.  Due to this, the paper does not meet the bar for acceptance.

**Reviewer Scores:**

It is likely that all reviewers may have maintained their scores.

---

### Decision · Program_Chairs · 2026-01-26

Reject